# SARS-CoV-2 tests, confirmed infections and COVID-19-related hospital admissions in children and young people: birth cohort study

Pia Hardelid [ID],[1] Graziella Favarato,[1] Linda Wijlaars,[1] Lynda Fenton,[2] Jim McMenamin,[3] Tom Clemens,[4] Chris Dibben,[4] Ai Milojevic,[5] Alison Macfarlane,[6] Jonathon Taylor,[7] Steven Cunningham,[8] Rachael Wood[2,9]

SC and RW are joint last authors.

**Correspondence to**
Dr Pia Hardelid; p.hardelid@ucl.ac.uk

## ABSTRACT

**Background** There have been no population-based studies of SARS-CoV-2 testing, PCR-confirmed infections and COVID-19-related hospital admissions across the full paediatric age range. We examine the epidemiology of SARS-CoV-2 in children and young people (CYP) aged <23 years.

**Methods** We used a birth cohort of all children born in Scotland since 1997, constructed via linkage between vital statistics, hospital records and SARS-CoV-2 surveillance data. We calculated risks of tests and PCR-confirmed infections per 1000 CYP-years between August and December 2020, and COVID-19-related hospital admissions per 100 000 CYP-years between February and December 2020. We used Poisson and Cox proportional hazards regression models to determine risk factors.

**Results** Among the 1 226 855 CYP in the cohort, there were 378 402 tests (a rate of 770.8/1000 CYP-years (95% CI 768.4 to 773.3)), 19 005 PCR-confirmed infections (179.4/1000 CYP-years (176.9 to 182.0)) and 346 admissions (29.4/100 000 CYP-years (26.3 to 32.8)). Infants had the highest COVID-19-related admission rates. The presence of chronic conditions, particularly multiple types of conditions, was strongly associated with COVID-19-related admissions across all ages. Overall, 49% of admitted CYP had at least one chronic condition recorded.

**Conclusions** Infants and CYP with chronic conditions are at highest risk of admission with COVID-19. Half of admitted CYP had chronic conditions. Studies examining COVID-19 vaccine effectiveness among children with chronic conditions and whether maternal vaccine during pregnancy prevents COVID-19 admissions in infants are urgently needed.

## WHAT IS ALREADY KNOWN ON THIS TOPIC

⇒ Children are less likely to suffer severe symptoms of SARS-CoV-2 infection than adults. There are few population-based studies of the epidemiology of SARS-CoV-2 in children not admitted to hospital.

## WHAT THIS STUDY ADDS

⇒ Using a national birth cohort from Scotland during 2020, we found that children and young people with chronic conditions were more likely to be tested, but secondary school-aged children with chronic conditions were less likely to have a confirmed infection. Infants and children/young people with chronic conditions were at highest risk of admission.

## HOW THIS STUDY MIGHT AFFECT RESEARCH, PRACTICE OR POLICY

⇒ Studies examining COVID-19 vaccine effectiveness among children with chronic conditions and whether maternal vaccine during pregnancy prevents COVID-19 admissions in infants are urgently needed.

## BACKGROUND

Children are much less likely to experience hospital admission and mortality related to SARS-CoV-2 infection than adults.[1] In Europe in 2020, 1.7% of COVID-19-related hospital admissions were in children <19 years of age.[2] Over the course of the pandemic, our understanding of how SARS-CoV-2 infection affects children has also improved. Children who experience more severe symptoms of

SARS-CoV-2 may present with acute infection symptoms such as fever or cough.[3–5] Other children may develop an acute inflammatory syndrome, paediatric inflammatory syndrome temporally associated with SARS-CoV-2 (PIMS-TS; also referred to as multisystem inflammatory syndrome related to COVID-19), several weeks after initial infection.[6–8] Children aged <2 years old appear to be over-represented among children admitted to hospital with acute symptoms, whereas children aged 10 years or older account for the largest proportion of admitted PIMS-TS cases.[4 9]

Among children admitted to hospital with SARS-CoV-2 or PIMS-TS, those with specific chronic respiratory, neurological, gastrointestinal or cardiovascular conditions, and particularly children with multiple comorbidities, were at increased risk of paediatric intensive care unit (PICU) admission or death.

Infants and teenagers appeared to have higher odds of these severe outcomes compared with children aged 1–4 years old.[10 11] A lower reported risk of severe disease and, until 2021, relatively lower rates of infection in children, have supported a narrative that the benefits and risks (primarily of myocarditis following second dose mRNA vaccines in young men[12 13]) of vaccinations in children are finely balanced.

Most studies of paediatric SARS-CoV-2 infection have been case series of infected or hospitalised children, making calculations of population-based risks of confirmed infections and associated admissions among different groups of children, including children with chronic conditions, impossible. Our aim was to provide population-based estimates of risk of SARS-CoV-2 testing, PCR-confirmed infections and COVID-19-related admissions in children and young people (CYP) based on age, presence of chronic conditions, and socioeconomic status during 2020 that could support vaccination and other policy recommendations across the population of CYP.

## METHODS
### Data sources

We used a national birth cohort of all CYP born in Scotland from 1997 onwards, developed from administrative health datasets linked to public health surveillance data on SARS-CoV-2 test results, originally constructed for the PICNIC Study.[14] Birth registrations comprised the cohort spine, and CYP are linked over time and between databases using the Community Health Index number, a unique personal identifier recorded at all interactions with the Scottish National Health Service. Table 1 summarises the databases and variables used in this study.

### Study population and follow-up

We included CYP born in Scotland from 1 April 1997 to 31 December 2020. Children born at less than 24 weeks' gestation or with a birth weight <500 g were excluded,[15] as were CYP whose mothers were not residents in Scotland at the time of delivery, and CYP who migrated out of Scotland before 1 February 2020. For analyses of SARS-CoV-2 tests and positive test results (from now on referred to as PCR-confirmed infections), CYP were followed from birth or 1 August 2020 (whichever occurred last), until death, migration from Scotland, their 23rd birthday or 31 December 2020, whichever occurred first. The date 1 August 2020 was chosen as the follow-up start date for analyses of tests and PCR-confirmed infections since this is when testing for SARS-CoV-2 became commonly available in the community (rather than solely in hospitals) for children of all ages.[16] Children, like adults, were advised to seek PCR testing if they developed a continuous cough, high temperature or loss of sense of smell or taste.[17 18] For calculation and analyses of rates of COVID-19-related admissions, we used 1 February 2020 as the follow-up start date. This allowed us to include all COVID-19-related hospital admissions since the start of the pandemic.

**Table 1** Datasets and variables from the national Scottish birth cohort used in the study

| Dataset | Dataset details | Variables used |
| --- | --- | --- |
| National Records for Scotland (NRS) birth registrations | Vital registration data on all children born in Scotland and their parents, collected via registry offices | Week and year of birth; baby sex; socioeconomic position (parents' occupation at birth) |
| Scottish Morbidity Record (SMR)-01 | Contains data on post-neonatal admissions and day cases to all NHS hospitals in Scotland | Admission and discharge dates; primary and secondary diagnoses during admission; type of hospital admission; admission and discharge data from intensive care unit |
| SMR-02 (maternity records) | Contains data on all maternity admissions (including deliveries) in Scotland | Estimated gestational age; birth weight; number of older siblings (parity) |
| COVID-19 tests | Contains data on all PCR and antigen tests for SARS-CoV-2 with results and dates | Date of testing; type of test; result |
| NRS death registrations | Vital registration data on children who died in Scotland | Date of death; cause of death |
| Scottish Birth Records | Contains data on all children born in NHS hospitals, with data on neonatal admissions in and after April 2003 | Diagnoses recorded at or shortly after birth; primary and secondary diagnoses at birth admission |
| Community Health Index (CHI) Register | Contains data on migration in/out Scotland | Migration outside Scotland |
| Child Health Surveillance Programme-School | Contains data on school health visits | Height and weight at age 5 |

NHS, National Health Service.

## Outcomes

Our primary outcomes were rates of SARS-CoV-2 PCR tests (positive or negative), PCR-confirmed SARS-CoV-2 infections and COVID-19-related hospital admissions. Our secondary outcomes were PIMS-TS admissions and COVID-19-related intensive care unit (ICU) stays. Online supplemental text 1 details how each of these outcomes was derived.

## Risk factors

We examined four risk key factors for testing, confirmed infections and hospital admission outcomes: age group, sex, family socioeconomic position and history of chronic conditions. Age as of 1 February 2020 was grouped into: <1 year (this also includes children born during 2020), 1–4 years, 5–11 years, 12–17 years and 18–22 years. We chose these age groups to reflect likely mixing patterns based on age (ie, prior to formal childcare, nursery/preschool, primary school, secondary school, and higher/further education or work). Family socioeconomic position was defined using parents' (father's or mother's if the birth was not jointly registered) occupation recorded on birth registration, coded using the UK National Statistics Socio-economic Classification (NS-SEC).[19] We collapsed the NS-SEC classes into: high (managerial and professional occupations), middle (intermediate occupations) and low (routine and manual occupations) socioeconomic position. We identified history of chronic conditions by examining International Classification of Disease version 10 (ICD-10) diagnostic codes recorded in the Scottish Morbidity Record (SMR-01) between 1 January 2015 and 31 January 2020, using an existing code list.[20] For children aged less than 5 years at the start of February 2020 or born during 2020, we used all available SMR-01 data and any diagnoses recorded on Scottish Birth Records (SBR). Chronic conditions were classified into eight types: developmental/mental health, blood/cancer, chronic infections, respiratory, metabolic/gastrointestinal/endocrine/genitourinary, musculoskeletal/skin, neurological/sensory, and cardiac conditions. These were further grouped into none, one type of condition and more than one type of chronic condition for analyses.

We further explored whether gestational age and the number of older siblings affected PCR-confirmed infection and hospital admission risk in children aged <5 years, and body mass index (BMI) in CYP aged 5–17 years. Gestational age was grouped as: preterm (<37 weeks) and term/late term (≥37 weeks). Number of older siblings (indicated by parity) was grouped as: no older siblings, one older sibling and two or more older siblings. BMI was derived from the Child Health Surveillance Programme-School dataset collected from children starting their first year at school (at age 5 years), and categorised[21] as underweight (<5th percentile), healthy weight (5th–<85th percentile) and overweight/obese (≥85th percentile).

## Statistical analyses

We calculated rates of testing and PCR-confirmed infections per 1000 CYP-years and hospital admission per 100 000 CYP-years with 95% CIs stratified by each risk factor. We estimated the median length of stay with interquartile ranges (IQRs) for COVID-19-related hospital admissions. We calculated the proportion of children with COVID-19 who had a chronic condition recorded either at baseline, or during the COVID-19 admission.

We examined the association between risk factors and testing rates using Poisson regression models with robust SEs to account for multiple tests per child. To examine the association between risk factors and PCR-confirmed SARS-CoV-2 infection, and COVID-19-related admission risk, we used Cox proportional hazards regression models. Where a child had multiple COVID-19-related admissions, only the first was included in the models. For each primary outcome, we first fitted an overall model including all ages and age group, sex, socioeconomic position and history of chronic conditions as risk factors. We tested for interaction with age group and each of the other main risk factors using the Wald test. Two-sided $p<0.05$ was considered statistically significant.

We then fitted models for each primary outcome stratified by age group if a statistically significant interaction with age was identified for any of the other variables or if we identified non-proportional hazards. In further analyses for ages <5 years old, we included parity and gestational age as additional risk factors in the models; and for ages 5–17 years, we included BMI category. We tested the proportional hazards assumption of the Cox model by inspecting plots of Schoenfeld residuals[22] and survival curves according to each main risk factor.

As there was only a small number of events for our secondary outcomes, we report the number of cases, median length of stay and age (with IQRs) only. All analyses were based on complete cases, as only a small number of CYP were missing values for any of the main variables. All statistical analyses were performed using Stata V.16.0.

## Sensitivity analyses

We examined the number of COVID-19 hospital admissions that were identified as occurring up to 14 days after a positive SARS-CoV-2 test. We repeated the analyses for hospital admission risk using a more specific definition of a COVID-19-related admission restricted to emergency admissions with an ICD-10 code indicating COVID-19 (U07.1 or U07.2)[23] as the primary diagnosis.

## Patient and public involvement

The PICNIC Study has been presented to a number of parent groups, including the Great Ormond Street Hospital Biomedical Research Centre Parents and Carers Advisory Group, and a coffee morning for parents at Shelter's Birmingham Office. This COVID-19 epidemiology substudy has not been specifically reviewed by parents.

 

## RESULTS

This study included 1 226 855 CYP (online supplemental figure 1). The median age in February 2020 was 11 years (IQR 5–17), and 8.0% of the cohort (97 884 of 1 226 855 CYP) had at least one chronic condition recorded in their hospital or birth record in the previous 5 years (online supplemental table 1).

### SARS-CoV-2 testing

Between 1 August (week 31) and 31 December 2020 (week 52), we identified 378 402 PCR tests linked to 256 741 CYP; 20.9% of CYP in the cohort had at least one test. Online supplemental figure 2 shows the weekly number of PCR tests by age group. The crude testing rate was 770.8 (95% CI 768.4 to 773.3) per 1000 CYP-years. The majority of CYP had been tested only once (200 288; 78.0%); 40 188 (15.7%) had been tested twice and 16 265 (6.3%) more than twice. Further results regarding rates of testing by week and risk factor can be found in online supplemental text 2 and online supplemental tables 2–5.

### PCR-confirmed infections

Among the 378 402 PCR tests identified in the cohort, 20 003 (5.3%) were positive and 7275 (1.9%) were void. Excluding multiple positive tests per CYP, this corresponds to 19 005 PCR-confirmed index infections in 7.4% (19 005 of 256 741) of the CYP who were tested between 1 August 2020 and 31 December 2020.

The overall rate of PCR-confirmed infections was 179.4 (95% CI 176.9 to 182.0) per 1000 CYP-years. Young adults (aged 18–22 years) had the highest rates of PCR-confirmed infections and those aged 1–4 years the lowest (table 2). Infants had the highest PCR-confirmed infection rates among preschool children, otherwise infection rates were positively correlated with age. CYP with chronic conditions had a lower risk of PCR-confirmed infection, particularly among secondary school-aged children

(online supplemental table 4 and table 3). Age group-specific analyses showed that among preschool children, PCR-confirmed infection rates were higher among children from lower socioeconomic backgrounds, whereas the opposite was observed among CYP aged 12 years and above (table 3).

Children aged <5 years with one older sibling had a reduced risk of a PCR-confirmed infection compared with children with no older siblings (online supplemental tables 6 and 7). Further, in children aged 12–17 years, being overweight/obese increased the risk of a PCR-confirmed infection compared with being of normal BMI.

### COVID-19-related hospital admissions

Between 1 February 2020 and 31 December 2020, there were 81 312 admissions in 55 940 CYP. Three hundred forty-six (0.6%) admissions in 318 CYP were identified as COVID-19 related. The median length of stay was 2 days (IQR 1–4 days). There were 110 admissions between February and July (31.8%) and 236 (68.2%) between August and December (figure 1). A total of 49.4% (n=157) of the 318 CYP admitted had at least one type of chronic condition recorded; and 23.3% (74 of 318; 46.5% of the 159 children with at least one chronic condition) had multiple types of chronic conditions recorded.

The overall COVID-19-related admission rate was 29.4/100 000 (95% CI 26.3 to 32.8) CYP years (table 4). Infants had the highest COVID-19-related admission rate: 120.6/100 000 (95% CI 92.2 to 157.9). CYP with chronic conditions, and particularly children with more than one chronic condition type recorded, had the highest admission rates across all age groups.

Of the CYP with chronic conditions who had a COVID-19-related admission, neurological/sensory conditions were the common condition type recorded among

**Table 2** Rates of PCR-confirmed infections (per 1000 CYP-years) by age group, sex, socioeconomic position and history of chronic conditions

| | Age <1 year | | | Age 1–4 years | | | Age 5–11 years | | | Age 12–17 years | | | Age 18–22 years | | |
|---|---|---|---|---|---|---|---|---|---|---|---|---|---|---|---|
| | Events | Rate | 95% CI | Events | Rate | 95% CI | Events | Rate | 95% CI | Events | Rate | 95% CI | Events | Rate | 95% CI |
| | 223 | 80 | 70 to 91 | 1136 | 62 | 58 to 65 | 3039 | 96 | 93 to 100 | 4929 | 193 | 188 to 198 | 9687 | 350 | 344 to 358 |
| Sex | | | | | | | | | | | | | | | |
| Male | 121 | 79 | 66 to 94 | 595 | 60 | 55 to 65 | 1545 | 91 | 87 to 96 | 2313 | 179 | 172 to 187 | 4487 | 365 | 355 to 376 |
| Female | 102 | 81 | 67 to 98 | 541 | 64 | 59 to 69 | 1494 | 102 | 97 to 108 | 2616 | 207 | 199 to 215 | 5200 | 338 | 329 to 348 |
| Socioeconomic position | | | | | | | | | | | | | | | |
| High | 21 | 51 | 33 to 78 | 141 | 51 | 43 to 60 | 292 | 81 | 73 to 91 | 503 | 175 | 161 to 191 | 1181 | 481 | 455 to 510 |
| Middle | 111 | 83 | 69 to 100 | 571 | 63 | 58 to 69 | 1446 | 102 | 97 to 108 | 2231 | 202 | 194 to 210 | 5510 | 353 | 344 to 363 |
| Low | 91 | 88 | 71 to 108 | 424 | 64 | 58 to 70 | 1301 | 94 | 89 to 100 | 2195 | 189 | 181 to 197 | 2996 | 312 | 301 to 324 |
| Chronic conditions | | | | | | | | | | | | | | | |
| None | 202 | 81 | 71 to 94 | 1031 | 63 | 60 to 67 | 2757 | 98 | 94 to 102 | 4591 | 199 | 194 to 205 | 8766 | 366 | 358 to 373 |
| 1 | 10 | 44 | 24 to 82 | 77 | 45 | 36 to 57 | 225 | 87 | 76 to 99 | 272 | 143 | 127 to 161 | 735 | 267 | 249 to 287 |
| >1 | 11 | 130 | 72 to 235 | 28 | 54 | 37 to 78 | 57 | 74 | 57 to 95 | 66 | 107 | 84 to 136 | 186 | 202 | 175 to 233 |

CYP, children and young people.

**Table 3** Time to PCR-confirmed infection: HRs by age group mutually adjusted for sex, socioeconomic position and history of chronic conditions

| | Age <1 years | | Age 1–4 years | | Age 5–11 years | | Age 12–17 years | | Age 18–22 years | |
|---|---|---|---|---|---|---|---|---|---|---|
| Number of CYP in model | 9661 | | 49 288 | | 70 245 | | 76 262 | | 70 212 | |
| Number of PCR-confirmed infections | 276 | | 1114 | | 1286 | | 2706 | | 4338 | |
| | **Adj HR** | **95% CI** | **Adj HR** | **95% CI** | **Adj HR** | **95% CI** | **Adj HR** | **95% CI** | **Adj HR** | **95% CI** |
| Sex | | | | | | | | | | |
| Male | 1 | – | 1 | – | 1 | – | 1 | – | 1 | – |
| Female | 1.15 | 0.93 to 1.41 | 1.08 | 0.97 to 1.20 | 1.10 | 1.02 to 1.19 | 1.20 | 1.15 to 1.27 | 0.96 | 0.92 to 1.00 |
| Socioeconomic position | | | | | | | | | | |
| High | 1 | – | 1 | – | 1 | – | 1 | – | 1 | – |
| Middle | 1.44 | 1.02 to 2.04 | 1.30 | 1.09 to 1.55 | 1.26 | 1.10 to 1.44 | 1.02 | 0.94 to 1.11 | 0.75 | 0.70 to 0.80 |
| Low | 1.46 | 1.02 to 2.08 | 1.31 | 1.10 to 1.58 | 1.17 | 1.02 to 1.34 | 0.91 | 0.83 to 0.98 | 0.66 | 0.62 to 0.71 |
| Chronic conditions | | | | | | | | | | |
| None | 1 | – | 1 | – | 1 | – | 1 | – | 1 | – |
| 1 | 0.62 | 0.39 to 0.98 | 0.74 | 0.60 to 0.91 | 0.87 | 0.75 to 1.00 | 0.79 | 0.71 to 0.88 | 0.76 | 0.70 to 0.82 |
| >1 | 1.23 | 0.72 to 2.10 | 0.79 | 0.55 to 1.12 | 0.76 | 0.58 to 1.00 | 0.59 | 0.48 to 0.73 | 0.58 | 0.50 to 0.67 |

CYP, children and young people; Adj HR, adjusted HR.

children aged <12 years, whereas among those aged 12–22 years, the most common conditions were developmental/mental health conditions.

Presence of one or more chronic conditions significantly increased the risk of COVID-19-related admissions across all ages (online supplemental table 4). In age-stratified analyses, presence of a chronic condition remained the only statistically significant risk factor for COVID-19-related admission across all age groups (table 5). We did not identify any statistically significant associations between prematurity, number of older siblings, or BMI category and COVID-19-related hospital

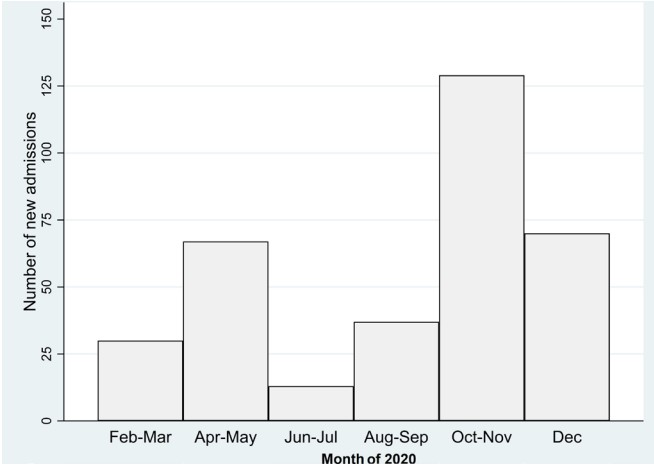

**Figure 1** Monthly number of COVID-19-related hospital admissions (weeks 5–52, year 2020; 1 February 2020–31 December 2020).

admission risk (online supplemental table 9); however, the number of hospital admissions was low in this study.

### ICU admissions and PIMS-TS cases

Thirteen (3.8%) of the 346 COVID-19-related admissions involved an ICU attendance, accounting for 1.2% of the 1238 ICU admissions in CYP during the study period. The vast majority of these admissions were in CYP with a history of one or more types of chronic conditions. The median age of CYP admitted to ICU was 14 years (IQR 9–19 years) and the median length of stay at ICU was 6 days (IQR 2–7 days).

We identified fewer than five admissions with a diagnosis suggestive of PIMS-TS and temporally associated with a positive PCR test (<28 days prior admission by definition), all in boys with an age spanning from 9 to 14 years. The median length of stay at admission was 10 days (IQR 6–14).

### Sensitivity analyses

Of the 346 COVID-19-related admissions, 203 (58.7%) had a specific COVID-19 ICD-10 code as the primary diagnosis and 258 (74.6%) were temporally associated with a SARS-CoV-2 positive test (figure 2). Using the more specific definition of a COVID-19-related admission, the admission rates were 107.0 (95% CI 80.4 to 142.4), 13.4 (9.0 to 19.8), 8.0 (5.6 to 11.7), 9.3 (6.3 to 13.6) and 27.7 (21.6 to 35.5) per 100 000 CYP-years in age groups <1, 1–4, 5–11, 12–17 and 18–22 years, respectively (online supplemental tables 10 and 11). This was between 12% (in infants) and 50% (in children aged 5–11 years) lower than the more inclusive definition used

**Table 4** Rates of COVID-19-related admissions (per 100 000 CYP-years) by age group, sex, socioeconomic position and history of chronic conditions

| | Age <1 year | | | Age 1–4 years | | | Age 5–11 years | | | Age 12–17 years | | | Age 18–22 years | | |
|---|---|---|---|---|---|---|---|---|---|---|---|---|---|---|---|
| | Events | Rate | 95% CI | Events | Rate | 95% CI | Events | Rate | 95% CI | Events | Rate | 95% CI | Events | Rate | 95% CI |
| | 53 | 121 | 92 to 158 | 51 | 27 | 21 to 36 | 55 | 16 | 12 to 21 | 49 | 18 | 13 to 23 | 110 | 49 | 41 to 67 |
| **Sex** | | | | | | | | | | | | | | | |
| Male | 30 | 133 | 93 to 190 | 22 | 23 | 15 to 35 | 31 | 17 | 12 to 25 | 26 | 18 | 12 to 27 | 43 | 38 | 28 to 55 |
| Female | 23 | 107 | 71 to 162 | 29 | 32 | 22 to 46 | 24 | 14 | 9 to 21 | 23 | 17 | 11 to 25 | 67 | 61 | 48 to 91 |
| **Socioeconomic position** | | | | | | | | | | | | | | | |
| High | * | 73 | 27 to 194 | * | 18 | 7 to 48 | * | 12 | 5 to 29 | * | 9 | 3 to 27 | 6 | 34 | 15 to 75 |
| Middle | * | 118 | 80 to 175 | * | 24 | 16 to 36 | * | 15 | 10 to 22 | * | 18 | 12 to 28 | 55 | 43 | 33 to 69 |
| Low | * | 139 | 93 to 207 | * | 34 | 23 to 50 | * | 18 | 12 to 26 | * | 19 | 13 to 29 | 49 | 62 | 47 to 86 |
| **Chronic conditions** | | | | | | | | | | | | | | | |
| None | * | 105 | 78 to 142 | * | 20 | 15 to 29 | 21 | 7 | 4 to 10 | 19 | 7 | 5 to 11 | 42 | 21 | 16 to 28 |
| 1 | * | 277 | 115 to 666 | * | 38 | 16 to 92 | 19 | 93 | 59 to 146 | 17 | 113 | 70 to 181 | 37 | 197 | 143 to 272 |
| >1 | * | 1032 | 387 to 2749 | * | 374 | 207 to 675 | 15 | 326 | 196 to 540 | 13 | 332 | 193 to 573 | 31 | 547 | 385 to 778 |

*Redacted due to small numbers in some groups.
CYP, children and young people.

in the main analyses. Presence of one or more chronic conditions remained the only significant risk factor for hospital admission with the specific definition (online supplemental tables 12 and 13) across the age groups. The median length of stay remained 2 days (IQR 1–5).

## DISCUSSION

Over one-fifth of CYP in Scotland had at least one SARS-CoV-2 PCR test during 2020, and 1.5% had a PCR-confirmed infection. CYP with chronic conditions were more likely to be tested, but secondary school-aged CYP with chronic conditions were less likely to have a PCR-confirmed infection. While COVID-19-related hospital admissions were uncommon (less than 3 per 10 000 CYP admitted in 2020), infants and CYP with chronic conditions recorded had the highest COVID-19-related admission rates.

The well-established Scottish data linkage infrastructure allowed us to include data for all CYP born in Scotland since 1997, thereby minimising selection bias and

**Table 5** Time to COVID-19-related admissions: HRs by age group mutually adjusted for sex, socioeconomic position and history of chronic conditions

| | Age <1 year | | Age 1–4 years | | Age 5–11 years | | Age 12–17 years | | Age 18–22 years | |
|---|---|---|---|---|---|---|---|---|---|---|
| Number of CYP in model | 92 530 | | 251 884 | | 347 542 | | 385 664 | | 268 467 | |
| N admissions | 53 | | 51 | | 55 | | 49 | | 110 | |
| | Adj HR | 95% CI | Adj HR | 95% CI | Adj HR | 95% CI | Adj HR | 95% CI | Adj HR | 95% CI |
| **Sex** | | | | | | | | | | |
| Male | 1 | – | 1 | – | 1 | – | 1 | – | 1 | – |
| Female | 0.82 | 0.51 to 1.30 | 1.49 | 0.88 to 2.51 | 1.10 | 0.62 to 1.96 | 1.05 | 0.67 to 1.67 | 1.47 | 0.99 to 2.17 |
| **Socioeconomic position** | | | | | | | | | | |
| High | 1 | – | 1 | – | 1 | – | 1 | – | 1 | – |
| Middle | 1.70 | 0.66 to 4.36 | 1.39 | 0.48 to 4.03 | 0.91 | 0.34 to 2.41 | 3.05 | 0.94 to 9.92 | 1.03 | 0.44 to 2.39 |
| Low | 2.07 | 0.81 to 5.28 | 1.97 | 0.69 to 5.61 | 0.92 | 0.35 to 2.43 | 2.75 | 0.85 to 8.93 | 1.27 | 0.54 to 2.99 |
| **Chronic conditions** | | | | | | | | | | |
| None | 1 | – | 1 | – | 1 | – | 1 | – | 1 | – |
| One | 3.14 | 1.50 to 6.58 | 2.46 | 1.10 to 5.53 | 13.73 | 6.99 to 26.96 | 13.85 | 8.12 to 23.61 | 8.86 | 5.62 to 13.96 |
| More than one | 10.99 | 4.74 to 25.48 | 19.71 | 10.45 to 37.19 | 49.81 | 24.30 to 102.12 | 40.96 | 22.89 to 73.30 | 25.39 | 15.80 to 40.82 |

CYP, children and young people; Adj HR, adjusted HR.

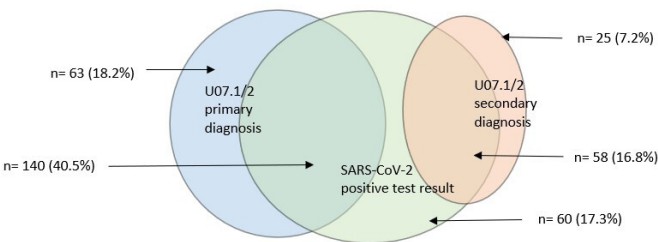

n= 63 (18.2%)

U07.1/2 primary diagnosis

SARS-CoV-2 positive test result

U07.1/2 secondary diagnosis

n= 25 (7.2%)

n= 140 (40.5%)

n= 58 (16.8%)

n= 60 (17.3%)

**Figure 2** Number of COVID-19-related admissions temporally associated with PCR-positive test up to 14 days before admission and by primary and secondary COVID-19 diagnosis (U07.1/U07.2) total admissions (n)=346.

loss to follow-up. We relied on linkage between hospital admission and public health surveillance data to define COVID-19-related admissions, allowing us to examine the robustness of our definitions in the linked data, rather than only relying on time difference between SARS-CoV-2 positive test and hospital admission alone. Indeed, we demonstrated that using a more specific definition of a COVID-19-related hospital admission (including only emergency hospital admissions with a specific COVID-19 ICD-10 code recorded as the primary diagnosis) decreased the rates by up to 50% compared with using a more sensitive definition based on either a recorded diagnosis or a positive SARS-CoV-2 PCR test up to 14 days before, or during, admission. We therefore recommend varying the definition of a COVID-19-related hospital admission via sensitivity analyses in studies using linked administrative data to ensure robustness of findings. By using a national birth cohort for this study, we examined variations in population-based rates of SARS-CoV-2 testing, PCR-confirmed infections and COVID-19-related hospital admissions across the full CYP age range, rather than only examining risk factors for ICU admission and death in hospitalised children.[24]

This study included data from the first year of the pandemic, when wildtype (until November 2020), followed by Alpha (dominant from December 2020) SARS-CoV-2 variants were circulating in Scotland. Our aim was to examine risk factors for SARS-CoV-2 testing, PCR-positive tests and COVID-19-related hospital admissions during 2020, rather than according to variant, given that the wildtype variant was circulating for the majority of the time period. This study will need to be repeated to examine the impact of later circulating variants, including Delta and Omicron, and changing transmission dynamics as vaccination of adults and reopening of schools, nurseries and workplaces since 2021 appear to be concentrating virus circulation among younger age groups.[25] The UK roll-out of COVID-19 vaccination for children aged 12–15 years started in July 2021,[26] and children aged 5–11 years from February 2022[27] is likely to change risks of hospital admission, particularly among children with chronic conditions. Further studies will need to examine whether the COVID-19 vaccination programme has amended the admission risks reported in this study. The results reported here provide baseline

risks during the first pandemic year against which more recent data can be compared. Updates of our analyses are planned.

We based our classification of chronic conditions on coded information in SMR-01 and SBR and may have missed some common conditions that are primarily managed in primary or community care settings, such as asthma and diabetes. Further, as we limited our lookback period to identify chronic conditions to 5 years, in order to avoid including conditions that may have resolved among older children, this may further have led to underascertainment of chronic conditions. Despite the use of a national birth cohort, the number of children admitted to hospital with SARS-CoV-2 during 2020 was small; therefore, we were unable to estimate admission risks in groups of children with specific chronic conditions. Our classification of socioeconomic position was based on parental occupation derived from birth certificates, which may not reflect current socioeconomic circumstances (eg, in older CYP).

As this study was based on linked, routinely collected data from the Scottish SARS-CoV-2 surveillance programme, our analyses of PCR-positive results relied on CYP (or in the case of younger children, their parents) coming forward for testing. Testing was recommended in individuals with high temperature, continuous cough or a loss of taste or smell; children are less likely than adults to display these symptoms when infected with SARS-CoV-2.[28] Further, our results regarding PCR-confirmed infections need to be interpreted in the context of testing behaviour. We demonstrated that during 2020, testing was more common among higher socioeconomic groups in preschool children, whereas in children aged 12 years and over, the lowest socioeconomic groups were more likely to test. These differences in testing behaviour are likely explained by factors such as presence of infection, severity and duration of symptoms, accessibility of testing and implications of test results for work, school and childcare.[29]

Infants had the highest admission risk. A systematic review has indicated that infants are also at highest risk of requiring PICU admission once in hospital with COVID-19.[30] However, admission rates in infancy related to SARS-CoV-2 (1/1000 child-years) during 2020 were lower than admission rates associated with confirmed influenza (2/1000 child-years)[31] or respiratory syncytial virus infections (22/1000 child-years).[32] Future research should examine how COVID-19 vaccination programmes for pregnant women and older children, and removal of non-pharmaceutical interventions to control population mixing, affect infant SARS-CoV-2 admission rates.

We demonstrated that a history of chronic conditions, particularly living with multiple different types of chronic conditions, was the most prominent risk factor for COVID-19-related hospital admission rates among CYP. Further, CYP with chronic conditions were more likely to be tested than those without, but less likely to

have a PCR-confirmed infection. This may reflect lower threshold for testing among high-risk groups.

Preschool and primary school children from lower socioeconomic backgrounds had higher risks of PCR-confirmed infection than children from higher socioeconomic backgrounds. Younger children spend more time in the home with their parents, thus their risk of infection is therefore more strongly associated with their parents' occupation (and ability to work from home). In older CYP, we instead identified higher PCR-confirmed infection rates among children of higher socioeconomic position, despite lower testing rates. This may be due to CYP from lower socioeconomic position groups being less likely to attend post-16 education, including university. There were large outbreaks in universities in Scotland in the autumn of 2020, which led to a surge in case numbers in those aged 18–22 years old.[33] Linkage between SARS-CoV-2 test results, hospital admission and education data is required to confirm whether exposure in education settings can explain these differences in infection risk.

We did not find a statistically significant association between socioeconomic position and risk of COVID-19-related admission. This is unlike some previous reports which have demonstrated higher all-age hospital admission rates in areas with higher area-level deprivation scores.[34] However, across all ages, the vast majority of COVID-19-related admissions are in adults. As COVID-19-related admission rates in children are much lower than in adults, systematic differences in admission rates by socioeconomic position among specific age groups of CYP are harder to detect, even when using national data. Further, as we used parental occupation to indicate socioeconomic background, this may not reflect current socioeconomic circumstances, as discussed above.

Our results showing that COVID-19-related admission rates in CYP peak in infancy indicate that further research and efforts to prevent COVID-19 admissions in children should include a focus on this age group. Pregnant women in Scotland are recommended to receive two doses of Pfizer/BioNTech COVID-19 vaccine.[35] As for pertussis[36 37] and influenza,[38 39] maternal vaccination during pregnancy could protect young babies from SARS-CoV-2 infection; however, no studies to date have examined this. Further, given that CYP with chronic conditions are more likely to be admitted to hospital admission with COVID-19 than other CYP, studies monitoring the effectiveness of COVID-19 vaccines against severe outcomes in these high-risk groups are required to determine whether vaccination reduces the risk of admission.

We identified a peak in COVID-19-related hospital admissions in infants, and presence of chronic conditions as the strongest risk factor for hospital admissions in CYP, yet half of CYP admitted did not have any chronic conditions recorded. Further studies are urgently needed to examine whether maternal vaccine during pregnancy prevents COVID-19 admissions in infants. These data also provide baseline risks of infection and hospital admission for risk–benefit assessments of childhood vaccination, particularly for preschool children.

**Author affiliations**
[1]Population, Policy & Practice Research and Teaching Department, University College London Great Ormond Street Institute of Child Health, London, UK
[2]Clinical and Public Health Intelligence Team, Public Health Scotland, Edinburgh, UK
[3]Respiratory Infection Team, Public Health Scotland, Edinburgh, UK
[4]School of Geosciences, The University of Edinburgh, Edinburgh, UK
[5]Department of Public Health, Environments and Society, London School of Hygiene & Tropical Medicine, London, UK
[6]Department of Midwifery and Radiography, City University of London, London, UK
[7]Faculty of Built Environment, Tampere University, Tampere, Finland
[8]Centre for Inflammation Research, University of Edinburgh, Edinburgh, UK
[9]Centre for Brain Sciences, University of Edinburgh, Edinburgh, UK

**Acknowledgements** We are grateful to Professor Bianca De Stavola (UCL) for her advice on statistical modelling. The authors would like to acknowledge the support of the eDRIS Team (Public Health Scotland), and particularly Diane Rennie, for their involvement in obtaining approvals, provisioning and linking data and the use of the secure analytical platform within the National Safe Haven. This work uses data provided by patients and collected by the NHS as part of their care and support.

**Contributors** PH conceived the study together with RW and SC. PH acquired the data supported by RW. GF and LW analysed the data. PH, GF and LW drafted the paper. All other authors read and commented on the paper. All authors have read and approved the final version. PH is the guarantor.

**Funding** This work was supported by UKRI-Medical Research Council (grant number MR/T016558/1).

**Competing interests** None declared.

**Patient and public involvement** Patients and/or the public were not involved in the design, or conduct, or reporting, or dissemination plans of this research.

**Patient consent for publication** Not required.

**Ethics approval** This study involves human participants and was approved by the University of Edinburgh School of Geosciences Ethics Committee (reference number 2020-401) and the Public Benefit and Privacy Panel for Health and Social Care (reference 1819-0049). This study involved the analysis of routinely collected NHS data that were not specifically collected for research purposes. As researchers did not have access to identifiable data, we could not ask the participants for consent.

**Provenance and peer review** Not commissioned; externally peer reviewed.

**Data availability statement** Data may be obtained from a third party and are not publicly available. Data are not publicly available. Researchers interested in using the data should apply to the Public Health Scotland eDRIS Team (https://www.isdscotland.org/products-and-services/edris/).

**ORCID iD**
Pia Hardelid http://orcid.org/0000-0002-0154-1306

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
