## [Reviewer comments · BMJ Paediatrics Open]

ARTICLE DETAILS

TITLE (PROVISIONAL)	SARS-CoV-2 tests, confirmed infections and COVID-19 related hospital admissions in children and young people: birth cohort study
AUTHORS	Hardelid, Pia Favarato, Graziella Wijlaars, Linda Fenton, Lynda McMenamin, Jim Clemens, Tom Dibben, Chris Milojevic, Ai Macfarlane, Alison Taylor, Jonathon Cunningham, Steven Wood, Rachael

VERSION 1 – REVIEW

REVIEWER	Reviewer name: Lorna Fraser Institution and Country: University of York, United Kingdom of Great Britain and Northern Ireland Competing interests: None
REVIEW RETURNED	01-Jun-2022

GENERAL COMMENTS	The authors have provided a nice analysis of a pre-existing dataset that should be published once the issues below have been addressed. Strengths Population based dataset Linkage to public health surveillance data Areas to address The inclusion of those up to age 22 is very confusing. You are legally an adult by age 16 in Scotland and compulsory education also ceases then. Given what we understand about the adult population and COVID I think the inclusion of this age range is confusing. Especially when SES is being applied based on parental factors and only secondary care data from the last 5 years are used for the ascertainment of chronic diseases. I therefore have concerns about misclassification bias in the young adults in this study and I would strongly suggest removing this age group.. Was adult and paediatric ICU data available for this study? Why was no PPI specific to this study undertaken? More comment on the lack of primary care data for the identification of chronic disease -eg moist children with asthma for example are treated in primary care. Also only using 5 years worth of hospital data - why and what about the implications of this? e.g even a child with a serious condition such as DMD may not have had a hospital admission in the last 5 years. There are vast amounts of supplemental data - ? all necessary
--

	I would have liked more discussion about the admission for covid vs the incidental finding of covid.
--	--

REVIEWER	Reviewer name: Dr. Christina Pagel Institution and Country: University College of London, United Kingdom of Great Britain and Northern Ireland Competing interests: None
REVIEW RETURNED	27-May-2022

GENERAL COMMENTS	SARS-CoV-2 tests, confirmed infections and COVID-19 related hospital admissions in children and young people: birth cohort study This paper provides a comprehensive overview of what happened to children in the first and start of the second wave of the pandemic in terms of test, confirmed PCRs and hospital admissions and as such is a valuable record. That said, I do not think the conclusion that this provides baseline evidence for risks of infection holds true, nor that it (much) informs risk-benefit for vaccination. I also think it would benefit from an explicit limitations / context section. Infections In terms of context, this study was conducted in the pre delta, pre omicron era (when the virus was less transmissible) and pre routine testing in schools (via lateral flow tests, which did not come into use widely until 2021). We know that the symptoms communicated to prompt PCR testing were less common in children, who are both more likely to be asymptomatic or paucisymptomatic *and* more likely to have atypical symptoms (e.g. gastro). This means children were significantly less likely to be tested than adults until routine school testing started in spring 2021. Thus the detailed testing (and test positive) data reported in this paper is interesting and valuable, but says as much about testing behaviours (on the part of the parents) more than it does about infections per se. Given that children who tested positive and their household members would need to isolate (by law), there was likely a socio-economic gradient in testing behaviours. That said, there was likely a socio-economic gradient going the other direction in exposure to infection. I don't see how this is possible to tease apart from the data but I think this needs to be discussed in the paper and acknowledged. Given that, I was surprised to see that the ONS infection data for Scotland was not mentioned – is this because age breakdown was not available for this data in 2020 in Scotland? I also found it odd to group rates of testing and positive test by age group but with no discussion of different exposure levels over time – covid was not a constant background exposure (e.g. I can't make sense of the "time to PCR infection" in table 3 – surely this depends on background pattern of covid & exposure??). For one thing, prevalence changed considerably over the course of 2020 *and* the measures in place to tackle it also changed. So lower infection rates during lockdowns would be expected and are not indicative of lower susceptibility but instead public health policy (and as such the learning going forward must surely be about the public health measures and overall prevalence as much as anything fundamental about child infection). The paper reads a bit as if these are estimates of fundamental risks in children from Covid as if the virus was a constant background that could also be projected in to the future and this was just not the case. I felt that the paper really
--

	would benefit from a discussion of changing prevalence and changing public health policy over the time period considered. Admissions In terms of hospital admissions, the definition is a little unclear. What is counted as a hospital covid-related admission? The sensitivity analysis talks about a positive test within 14 days of admission, but many CYP would only be tested *on* or *just after* admission with no community test. Are they counted as an admission or not? What about CYP with relevant ICD-10 codes but *no* recorded positive test? This is also particularly relevant for PIMS-TS – the authors mention a cut off of 28 days for attest (but isn't 42 days a more normal cut off?), but many children never had a positive test and are unlikely to test positive on admission for PIMS-TS. Was an ICD-10 code indicating PIMS-TS considered enough for inclusion? Overall Given that now we are in era of entirely different variants (with different virulence and transmissibility and immune escape), vaccination, extremely high previous infection and different (or no) public health measures, there is limited value in this data informing future course of this pandemic. BUT I still believe this is a valuable record of what did happen and should be published, but with these caveats in mind (they are partly discussed in the conclusions already). The most interesting bit to me is the detailed work identifying that about 50% of children in hospital had a pre-existing condition and the higher risk in under 5s. I believe the paper would benefit from revision emphasising more the fact that PCR infections measure test-seeking (and knowledge of symptoms!) as well as prevalence, and that all the rates take place in a changing background context (that continues to change). If possible I would also discuss the ONS Infection Survey and what says about prevalence in children (if age data is available for 2020).
--	--

REVIEWER	Reviewer name: Dr. Harish PEMDE Institution and Country: Kalawati Saran Children's Hospital, India Competing interests: None
REVIEW RETURNED	27-May-2022

GENERAL COMMENTS	This study provides population based risks of COVID19 infections, hospitalization and /risk factors' for hospitalization. This will prove a very important study to compare the risks and also to assess the benefits vaccination provides.
---

REVIEWER	Reviewer name: Dr. Jeremy Miles Institution and Country: Google Inc, United States Competing interests: None
REVIEW RETURNED	23-May-2022

GENERAL COMMENTS	This is an interesting and useful paper which exploits a dataset that is almost unique in the world. The paper is clearly written and analysis seems appropriate and is presented clearly. I have a small number of minor suggestions and comments. Thanks to the authors for placing the tables in the text. This makes my life much easier as reviewer. I find the first sentence of the results paragraph of the abstract a little hard to read. I'd prefer if the numbers that were relevant were next to each other, otherwise I need to read the sentence three times (at least). (i.e. "There were 378,402 tests, rates of rates of 770.8/1000 (95% confidence interval 768.4-773.3)). It's not clear to me why "hence" in the sentence "hence 20.9% of CYP in the cohort had at least one test". P17: "less than five" should be "fewer than five".
--

VERSION 1 – AUTHOR RESPONSE

Population, Policy & Practice Research and Teaching Department UCL Great Ormond Street Institute of Child Health 30 Guilford Street London WC1N 1EH 24th June 2022 Dear Editors, Re: Manuscript ID bmjpo-2022-001545 Thank you very much for your email of the 11th June inviting us to resubmit the paper with major revision. We have responded to the reviewers' queries below and attach the amended manuscript. We would like to thank the reviewers for their helpful comments which we believe substantially improved the quality of the manuscript. Please do not hesitate to get in touch if you have further queries. Yours sincerely, Pia Hardelid (on behalf of all co-authors)

Response to reviewers

Dr. Jeremy Miles, Google Inc This is an interesting and useful paper which exploits a dataset that is almost unique in the world. The paper is clearly written and analysis seems appropriate and is presented clearly. I have a small number of minor suggestions and comments. Thanks to the authors for placing the tables in the text. This makes my life much easier as reviewer. Thank you very much for these kind comments. I find the first sentence of the results paragraph of the abstract a little hard to read. I'd prefer if the numbers that were relevant were next to each other, otherwise I need to read the sentence three times (at least). (i.e. "There were 378,402 tests, rates of rates of 770.8/1000 (95% confidence interval 768.4-773.3)). Thank you for this suggestion, this has now been amended. It's not clear to me why "hence" in the sentence "hence 20.9% of CYP in the cohort had at least one test". We have now removed this word as we can see how it would be confusing. P17: "less than five" should be "fewer than five". Thank you this has now been changed.

Reviewer: 2 Dr. Harish PEMDE, Kalawati Saran Children's Hospital, Lady Hardinge Medical College

Comments to the Author This study provides population based risks of COVID19 infections, hospitalization and /risk factors' for hospitalization. This will prove a very important study to compare the risks and also to assess the benefits vaccination provides. Thank you for this comment.

Reviewer: 3 Dr. Christina Pagel, University College of London, Great Ormond Street Hospital NHS Foundation Trust

Comments to the Author SARS-CoV-2 tests, confirmed infections and COVID-19 related hospital admissions in children and young people: birth cohort study This paper provides a comprehensive overview of what happened to children in the first and start of the second wave of the pandemic in terms of test, confirmed PCRs and hospital admissions and as such is a valuable record. Thank you for this comment That said, I do not think the conclusion that this provides

baseline evidence for risks of infection holds true, nor that it (much) informs risk-benefit for vaccination. I also think it would benefit from an explicit limitations / context section. We have responded to each specific recommendation as below. In terms of context, this study was conducted in the pre delta, pre omicron era (when the virus was less transmissible) and pre routine testing in schools (via lateral flow tests, which did not come into use widely until 2021). We know that the symptoms communicated to prompt PCR testing were less common in children, who are both more likely to be asymptomatic or paucisymptomatic *and* more likely to have atypical symptoms (e.g. gastro). This means children were significantly less likely to be tested than adults until routine school testing started in spring 2021. Thus the detailed testing (and test positive) data reported in this paper is interesting and valuable, but says as much about testing behaviours (on the part of the parents) more than it does about infections per se. Given that children who tested positive and their household members would need to isolate (by law), there was likely a socio-economic gradient in testing behaviours. That said, there was likely to be a socioeconomic gradient going the other direction in exposure to infection. I don't see how this is possible to tease apart from the data but I think this needs to be discussed in the paper and acknowledged. As the reviewer has recognised, this is a study using linked data on SARS-CoV-2 tests carried out in hospitals and the community as part of the national testing programme. Therefore, the tests carried out would have been collected from CYP presenting in hospitals (throughout the study period), or as part of the Test, Trace, Isolate, Support (TTIS) Programme in Scotland (which was first implemented in May 2020; testing in the community became available for children of all ages from August 2020). Individuals, including children, with either continuous cough, a high temperature or a loss of sense or smell, were recommended to have a test. As we already present in the paper (supplementary tables 2 and 3), testing rates are indeed highest in the most affluent socio-economic group for children aged

VERSION 2 – REVIEW

REVIEWER	Reviewer name: Lorna Fraser Institution and Country: University of York, United Kingdom of Great Britain and Northern Ireland Competing interests: None
REVIEW RETURNED	01-Jun-2022
GENERAL COMMENTS	The authors have addressed most of my original comments - this paper is an important addition to the literature on covid. However I still find it confusing to be referring to 22 year olds as children rather than young adults. It would be helpful if the use of child and young person were more consistent throughout the paper.

REVIEWER	Reviewer name: Dr. Christina Pagel Institution and Country: University College of London, United Kingdom of Great Britain and Northern Ireland Competing interests: None
REVIEW RETURNED	27-May-2022
GENERAL COMMENTS	Thank you to the authors for addressing my comments so thoroughly. Also, apologies for missing the detailed inclusion criteria in the supplementary material! My concerns have all been dealt with and I recommend publication.

	Just 2 small things: 2 typos (page 8, line 41 - missing "hospital admission" i think; page 16, line 6 "ICU admission" not "ICU attendance"?) The finding that under 5s with no older siblings were more likely to test positive than those with 1 older sibling was interesting and counter-intuitive. AND the effect was not seen for >1 sibling. Do the authors have any idea why this is?
--	---